# Oxyntomodulin regulates resetting of the liver circadian clock by food

Dominic Landgraf[1†‡§], Anthony H Tsang[1,2†], Alexei Leliavski[1,2], Christiane E Koch[2], Johanna L Barclay[1], Daniel J Drucker[3,4], Henrik Oster[1,2]*

[1]Circadian Rhythms Group, Max Planck Institute for Biophysical Chemistry, Göttingen, Germany; [2]Chronophysiology Group, Medical Department I, University of Lübeck, Lübeck, Germany; [3]Department of Medicine, University of Toronto, Toronto, Canada; [4]Lunenfeld-Tanenbaum Research Institute, Mount Sinai Hospital, University of Toronto, Toronto, Canada

**Abstract** Circadian clocks coordinate 24-hr rhythms of behavior and physiology. In mammals, a master clock residing in the suprachiasmatic nucleus (SCN) is reset by the light–dark cycle, while timed food intake is a potent synchronizer of peripheral clocks such as the liver. Alterations in food intake rhythms can uncouple peripheral clocks from the SCN, resulting in internal desynchrony, which promotes obesity and metabolic disorders. Pancreas-derived hormones such as insulin and glucagon have been implicated in signaling mealtime to peripheral clocks. In this study, we identify a novel, more direct pathway of food-driven liver clock resetting involving oxyntomodulin (OXM). In mice, food intake stimulates OXM secretion from the gut, which resets liver transcription rhythms via induction of the core clock genes *Per1* and *2*. Inhibition of OXM signaling blocks food-mediated resetting of hepatocyte clocks. These data reveal a direct link between gastric filling with food and circadian rhythm phasing in metabolic tissues.

*For correspondence: henrik. oster@uksh.de

†These authors contributed equally to this work

Present address: ‡Department of Psychiatry, University of California, San Diego, San Diego, United States; §Center for Chronobiology, University of California, San Diego, San Diego, United States

Competing interests: The authors declare that no competing interests exist.

## Introduction

Extended night or rotating shift work is associated with an elevated risk for developing cancer, cardiovascular disease, immune deficiency, mood disorders, and metabolic alterations (*Rosenberg and Doghramji, 2011*; *Herichova, 2013*). One major factor believed to contribute to this adverse health impact of shift work is a disruption of endogenous circadian clocks by mistimed resetting stimuli, so called *Zeitgebers*, as a consequence of altered sleep/wake schedules. Most organisms have evolved internal timekeepers to anticipate the environmental changes brought about by the Earth's rotation around its axis. In mammals, these so called circadian clocks are based on ubiquitously expressed cellular interlocking transcriptional–translational feedback loops (TTLs) of clock genes/ proteins (*Albrecht, 2012*). In the core TTL, the transcriptional activators circadian locomotor output cycles kaput (CLOCK) and brain and muscle ARNT-like 1 (BMAL1; ARNTL) regulate the expression of two *Cryptochrome* (*Cry1/2*) and three *Period* (*Per1-3*) genes. Towards the end of the day, PER and CRY proteins translocate into the nucleus where they inhibit their own abundance via inhibition of CLOCK/BMAL1. Further accessory loops serve to stabilize this 24-hr feedback rhythm and integrate the clock with cellular processes (*Takahashi et al., 2008*). The clock machinery regulates physiology via orchestration of tissue-specific rhythmic expression of clock output genes (*Yan et al., 2008*).

The external light–dark cycle is the most prominent *Zeitgeber* of the central circadian pacemaker located in the suprachiasmatic nucleus (SCN) of the hypothalamus (*Golombek and Rosenstein, 2010*). The SCN receives light information from the retina and synchronizes peripheral clocks throughout the body via neuronal and hormonal pathways (*Dibner et al., 2010*). While the SCN itself is largely non-responsive to non-photic timing signals such as food intake, meal timing is an important

**eLife digest** Humans and other animals have adapted their behavior and their biology to the daily cycle of light and dark. Groups of genes are reliably switched on or off at different times of the day, and act as internal, or 'circadian', clocks that help these organisms to stay on a 24-hour cycle. External signals also synchronize the body's internal clocks. For example, sunlight helps synchronize the master clock in the brain, while mealtimes and other cues help other organs keep time.

These internal clocks are often disrupted in people who work overnight or on rotating shifts. It is believed that when these individuals wake up or go to sleep at odd times it confuses their circadian clocks, which can be harmful to their health. People who work these unusual hours are at an increased risk of developing cancer, heart disease, obesity, and other disorders that involve problems with metabolism.

Eating at odd hours may also throw off the circadian clocks in the digestive system. This may explain why metabolic problems have been linked to working odd hours. Landgraf, Tsang et al. hypothesized that if the hormones produced after eating are released when a person would normally be sleeping, this may desynchronize the circadian clock in organs like the liver. Screening mice and tissue samples from mice for hormones that perturb circadian rhythms showed that a hormone called oxyntomodulin, which is released from the gut after eating, activated important circadian clock genes in mouse livers. The increases in clock gene activation were comparable to those seen in the brain in response to exposure to light.

Landgraf, Tsang et al. revealed that the clock-resetting effects of oxyntomodulin were the greatest when animals were exposed to it by eating, or by injections of the hormone, at times when the animals would normally be fasting. The experiments also showed that blocking oxyntomodulin prevented eating at unusual times from interfering with the liver's circadian clocks. The findings may suggest a way to help protect people who work overnight from the harmful health effects linked to perturbed circadian clocks.

*Zeitgeber* for clocks in peripheral tissues (*Stokkan et al., 2001*). If food access is restricted to the normal rest phase of an organism, that is, the night for humans or daytime for nocturnal rodents, peripheral clocks become uncoupled from the SCN and adapt to the timing of food availability (*Damiola et al., 2000*). Shift workers often eat at times when their digestive timing system is poorly prepared for food (*Lowden et al., 2010*). Animal studies suggest that food intake during the normal rest phase promotes obesity (*Arble et al., 2009*; *Hatori et al., 2012*) and peripheral circadian uncoupling has been suggested to contribute to the development of metabolic disorders in night shift workers (*Antunes et al., 2010*; *Barclay et al., 2012*). Various other factors can regulate clock gene expression in peripheral tissues, including glucocorticoids and changes in body temperature or autonomic signaling (*Dibner et al., 2010*). The mechanisms of food-dependent peripheral clock resetting, however, remain poorly understood. Metabolic hormones such as insulin, ghrelin, and glucagon (GCG) have been shown to affect circadian rhythms associated with food restriction (*LeSauter et al., 2009*; *Tahara et al., 2011*; *Chaves et al., 2014*; *Sun et al., 2015*). While ghrelin appears to act primarily on the brain, insulin and GCG levels are mainly regulated via blood glucose. However, it was shown that carbohydrate intake alone has only a minor phase resetting capacity, while complex foods show much stronger effects (*Hirao et al., 2009*), indicating that other factors must be involved. Besides the pancreas, other organs—notably including the gastrointestinal tract itself—show acute hormonal responses to fasting or feeding (*Stanley et al., 2005*). This led us to hypothesize that postprandial, gut-derived signals may be implicated in food-driven resetting of peripheral clocks. In a screen using rhythmic liver slice cultures, we identified oxyntomodulin (OXM) as a potent resetting signal of liver circadian clocks. OXM is an anorexigenic incretin hormone produced in the gut by prohormone convertase 1/3-driven cleavage of the precursor preproglucagon (for review see *Drucker, 2005*). It modulates energy and glucose metabolism by acting on various tissues, including brain, liver, and pancreas (*Baldissera et al., 1988*; *Gros et al., 1993*). Since OXM secretion is dependent on food intake, we hypothesized that OXM may directly link food intake to hepatic transcriptional activity by resetting of the liver clock.

## Results

### OXM resets the circadian clock in organotypic liver slice cultures

We screened a commercially available metabolic peptide library (Obesity Peptide Library, Phoenix Europe GmbH; DE) for factors capable of resetting luciferase activity rhythms in organotypic liver slice cultures from *Per2::LUC* circadian reporter mice (*Yoo et al., 2004*). Interestingly, out of 200 peptides applied during the descending phase (~180°, corresponding to the early morning) of the luciferase activity rhythm, only a few produced marked phase shifts, including three proglucagon-derived peptide (PGDP) hormones: exendin-4, OXM, and GCG (*Figure 1—source data 1*). Exendin-4 has been isolated from the salivary gland of the *Gila* monster, with no analogue in rodents or humans. To compare the effectiveness of mammalian PGDPs in liver clock resetting, we treated slices with increasing doses of OXM, GCG, and the three other commercially available PGDPs, glicentin-related pancreatic polypeptide (GRPP), glucagon-like peptide-1 (GLP-1), and glucagon-like peptide-2 (GLP-2) (*Figure 1A*). GLP-1, GLP-2, and GRPP (0.5–450 nM) had no significant resetting effects on PER2::LUC phase compared to PBS-treatment (*Figure 1B–D*). GCG resulted in phase delays of up to 3 hr, but only at relatively high concentrations (*Figure 1E*). In contrast, OXM reset PER2::LUC rhythms in liver slices at much lower doses, resulting in phase delays of up to 8 hr at higher concentrations (*Figure 1F*). To act as a true *Zeitgeber* signal one would expect differential OXM effects depending on treatment time, that is, a circadian *gating* effect. We tested this by applying OXM at different phases of the PER2::LUC rhythm. To validate the setup, slices were treated at different PER2::LUC phases with 100 µM of the glucocorticoid analog dexamethasone (DEX), which was previously shown to reset hepatocyte clocks in vivo (*Balsalobre et al., 2000*). In a phase-dependent manner DEX treatment reset PER2::LUC activity rhythms in slices (*Figure 1—figure supplement 1*). Very similar to what had been observed after DEX treatment in animals (*Balsalobre et al., 2000*), application in the first quarter of the PER2::LUC activity rhythm (0–90°) resulted in phase delays, while later treatments produced phase advances (100–180°) or had no marked effect (around 270°). Likewise, OXM effects were phase dependent. Delays were predominantly observed at 90–210° of the PER2::LUC cycle with a maximum around 180°, while only modest phase shifts were seen at 270–360° (*Figure 1G*). Though GCG also showed potential in resetting liver clock rhythms, OXM emerged as the most potent liver clock synchronizer from our screen. Moreover, contrary to GCG, OXM secretion is directly induced by food consumption in humans (*Le Quellec et al., 1992*), making it an attractive candidate for linking meal timing and clock function. Therefore, we focused on OXM for further analyses.

### OXM signals via the GCG receptor to activate *Per* gene expression in liver slices

So far no OXM-specific receptor has been identified; however, OXM can bind to and activate both GCG and GLP-1 receptors (*Jorgensen et al., 2007*). *Gcgr* transcripts are strongly expressed in the murine liver (*Sinclair et al., 2008*). In contrast, the majority of previous studies failed to detect a full-length *Glp1r* mRNA in murine hepatocytes (*Campos et al., 1994*; *Dunphy et al., 1998*; *Panjwani et al., 2013*). We performed RT-PCR analyses for all annotated coding *Glp1r* exons on cDNA preparations from wild-type mouse livers with pancreas as positive control. *Glp1r* transcripts were present in pancreas, but undetectable in liver samples (*Figure 2—figure supplement 1*), in line with the absence of significant liver clock resetting effects of GLP-1 (*Figure 1B*) and potent resetting of PER2::LUC rhythms by OXM treatment in slices from *PER2::LUC x Glp1r$^{-/-}$* mice (*Figure 2A*). On the other hand, blocking glucagon receptors (GCGR) signaling by co-treatment with 2-(4-Pyridyl)-5-(4-chlorophenyl)-3-(5-bromo-2-propyloxyphenyl)pyrrol (Calbiochem Glucagon Receptor Inhibitor II; GRI-2) potently inhibited GCG- and OXM-induced clock resetting in *Per2::LUC* slices (*Figure 2B*).

GCGR is a G protein-coupled receptor that, via protein kinase A, leads to phosphorylation and activation of the transcription factor cyclic adenosine monophosphate (cAMP) response element-binding protein (CREB) (*Gonzalez and Montminy, 1989*; *Dalle et al., 2004*). This pathway is reminiscent of the SCN, where nocturnal light exposure induces *Per* gene transcription via cAMP signaling and CREB activation downstream of the N-methyl-D-aspartate receptor (*Welsh et al., 2010*). To investigate if OXM would impinge on the hepatic clock machinery in a similar way, we treated liver explants with OXM and performed chromatin immunoprecipitation (ChIP) analysis to measure CREB binding to cAMP response elements (*CRE*) in the *Per1* gene promoter. 30 min after

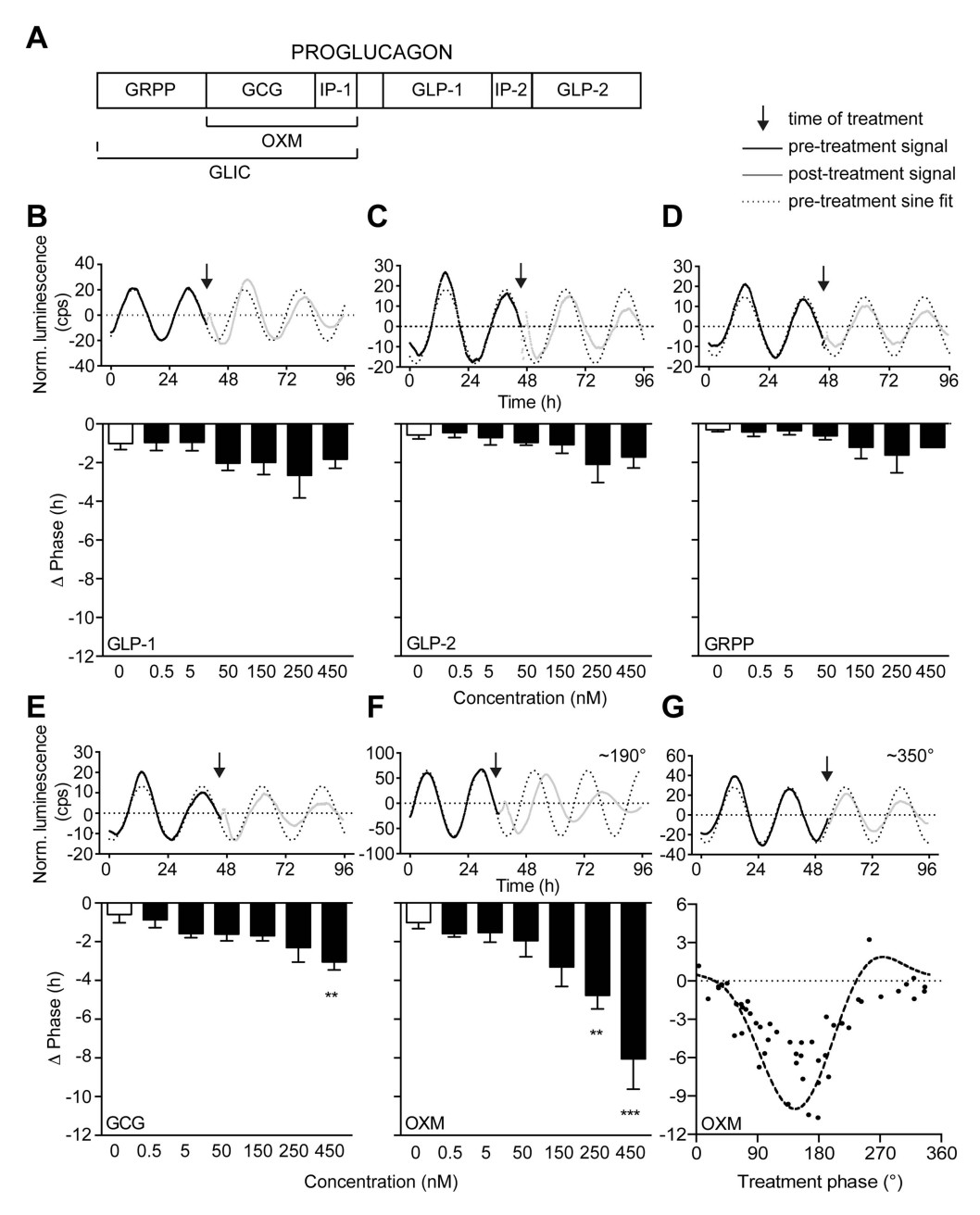

**Figure 1**. Oxyntomodulin (OXM) phase- and dose-dependently resets circadian clocks in liver slices. (**A**) Schematic sequence of the proglucagon-derived peptides (GRPP—glicentin-related pancreatic peptide; GLIC—glicentin; OXM—oxyntomodulin; GCG—glucagon; IP-1—intervening peptide-1; GLP-1—glucagon-like peptide-1; IP-2—intervening peptide-2; GLP-2—glucagon-like peptide-2). (**B**–**F**) Example luminescence traces and dose-dependent responses for GLP-1 (**B**; $F_{(6, 28)} = 1.509$), GLP-2 (**C**; $F_{(6, 28)} = 1.530$), GRPP (**D**; $F_{(6, 28)} = 1.151$), GCG (**E**; $F_{(6, 28)} = 3.569$), and OXM (**F**; $F_{(6, 28)} = 8.790$)-induced phase resetting of PER2::LUC rhythms in liver slices treated at 180–200°. Data are presented as mean ± S.E.M. (n = 5). One-way ANOVA (F-values with degrees of freedom provided in brackets): *$p < 0.05$; **$p < 0.01$; ***$p < 0.001$. Asterisks indicate significant differences relative to PBS treatment (white bars). (**G**) Phase response curve for OXM-induced phase resetting of PER2::LUC rhythms in liver slices. Circles: raw data of individual slices; dashed line: sine wave regression with harmonics.

*Figure 1. continued on next page*

*Figure 1. Continued*

The following source data and figure supplements are available for figure 1:

**Source data 1**. Table of effects of metabolic peptide treatment on PER2::LUC liver slice rhythms.

**Figure supplement 1**. Phase response curve for dexamethasone (DEX) treatment in *Per2::LUC* liver slice cultures.

OXM treatment, CREB binding was significantly increased at the *Per1* CRE, but not at downstream sequences (*Figure 2C*). In addition, we analyzed clock gene expression in liver slices at different intervals after treatment with OXM at 180°. *Per1* expression was transiently induced 60 min after addition of OXM to the medium, returning back to normal levels after 120 min (*Figure 3A*). Similarly, *Per2* was induced by OXM after 60 min, but mRNA levels remained high even after 120 min (*Figure 3B*). No significant effect was seen on *Bmal1* expression at all time points (*Figure 3C*). In line with the absence of OXM-induced phase shifts at 360° (*Figure 1G*), *Per1/2* and *Bmal1* mRNA levels were unaffected by OXM treatment at this phase (*Figure 3D–F*). Induction of *Per1* and *Per2* expression was preserved in *Glp1r*$^{-/-}$ slices, suggesting that hepatic OXM effects are independent of GLP-1R signaling (*Figure 3G,H*).

## OXM resets the liver clock in vivo

To test if the effect of OXM on liver clock gene activity is preserved in vivo, we analyzed hepatic *Per1/2* transcription after OXM treatment in wild-type mice. Analogous to what we observed in slices, robust *Per1* and *Per2* induction was observed after *i.v.* injection of OXM at *Zeitgeber* time (ZT) 3 (*Figure 4A*). When animals were treated with OXM at the opposite phase of the LD cycle (ZT15)—a time when nocturnal mice usually eat and, thus, no food-mediated clock shifts would be expected—no induction of *Per* gene expression was observed (*Figure 4B*). Of note, in situ hybridization (ISH) showed no acute effect of OXM treatment on *Per* expression in the SCN at ZT3 (*Figure 4C*), indicating that OXM acts primarily on the liver clock and in line with the observed phase stability of the SCN clock under time-restricted feeding conditions (*Damiola et al., 2000*).

To assess OXM effects on liver clock phase, we next treated wild-type mice with either PBS or OXM at the beginning of their rest phase on the first day in constant darkness (DD) and under fasting conditions, thus excluding potential confounding effects of light exposure or food intake. *Per2* and *Dbp* expression were determined from liver cDNA preparations at different time points using qPCR. We detected phase delays of *Per2* and *Dbp* mRNA rhythms in livers of OXM-treated mice relative to those in PBS-injected control animals (*Figure 4D*), in line with the phase-delaying effects of OXM administration in slices at this time (*Figure 1G*). In addition, *Per2*, but not *Dbp*, rhythms appeared dampened after OXM injection.

Tissue clocks regulate local physiology via coordination of transcriptional programs. To test if OXM treatment would impinge on hepatic energy metabolism, we analyzed the expression of important metabolic transcripts after OXM treatment. Similar to what we observed for clock gene activity, transcript profiles of genes involved in liver carbohydrate metabolism were found either phase delayed (*Foxo1* and *Pdk4*; *Figure 5A,B*) and/or dampened (*Foxo1* and *Pklr*; *Figure 5A,C*). Of note *Pepck*, which was previously described as a clock output gene in liver (*Lamia et al., 2008*) was not rhythmic under these conditions, and OXM had no further effect on *Pepck* mRNA levels (*Figure 5D*). Expression levels of the glucose transporter *Slc2a2* (*Glut2*) and the pyruvate transporter *Slc16a7* were also dampened or phase-delayed, respectively (*Figure 5E,F*).

In summary, our data so far show that OXM treatment resets liver circadian mRNA rhythms in a phase- and dose-dependent manner, indicating that it may be involved in food-induced resetting of the liver circadian clock and metabolic machinery.

## Food intake-mediated OXM induction and liver clock resetting

In humans, OXM levels in the blood rise in response to food intake (*Le Quellec et al., 1992*). To test whether this effect is conserved in mice, we determined diurnal plasma oxyntomodulin-like immunoreactivity (OLI) profiles in mice with ad libitum food access and in fasted animals. In fed mice, OLI levels were elevated during the active dark phase, while under fasting conditions OLI concentrations were constant and consistently low (*Figure 6A*). In line with this, non-rhythmic *Gcg*

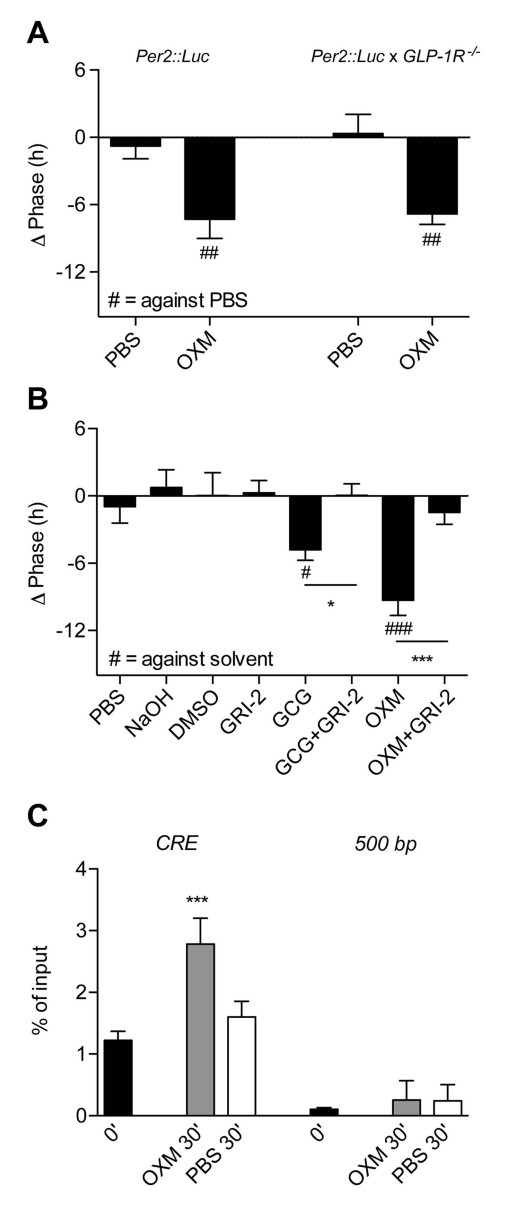

**Figure 2**. Glucagon (GCG) receptor regulates phase resetting effects of OXM and GCG in *Per2::LUC* liver slices. (**A**) OXM-induced phase shifts in *Per2::LUC* and *Per2::LUC* x *Glp1r*$^{-/-}$ liver slices. Mann–Whitney test: ##p < 0.01 against solvent. (**B**) GCG and OXM-induced phase shifts in *Per2::LUC* slices are abolished by co-treatment with GRI-2. One-way ANOVA with Bonferroni post-test: p < 0.05; ###p < 0.001 against solvent; *p < 0.05; ***p < 0.001. Data are presented as mean ± S.E.M. (n = 8); $F_{(7, 56)}$ = 7.314. (**C**) OXM treatment promotes binding of CREB to CRE elements at the *Per1* gene promoter. One-way ANOVA with Bonferroni post-test: ***p < 0.001 against 0'. Data are presented as mean ± S.E.M. (n = 5; $F_{(5, 24)}$ = 22.2).

The following figure supplement is available for figure 2:

**Figure supplement 1**. Absence of *Glp1r* transcripts in mouse liver.

mRNA expression in the gut was observed under fasting conditions (data not shown). These data suggest a link between food intake and OXM secretion. To test this more directly, we used a 12-hr fasting-refeeding paradigm. Mean fasting OLI levels in the early morning (ZT1) were ~3.5 ng/ml, but showed high inter-individual variation (*Figure 6B*). Upon food intake, a rapid increase (relative to individual fasting levels) was observed after 20 min. This effect persisted for more than 1 hr before returning to baseline levels (*Figure 6C*). Of note, OLI induction after OXM injections were about twofold–threefold higher than what was observed after refeeding (*Figure 6—figure supplement 1*). To test if postprandial OLI induction is sufficient to affect liver clock gene expression, we analyzed *Per1/2* mRNA levels in livers of wild-type mice after fasting-refeeding. Parallel to the rise in plasma OLI, we observed a transient postprandial increase of hepatic *Per1* expression. *Per2* expression showed a delayed, but a more persistent induction (*Figure 6D*). Food-mediated *Per* activation was partly inhibited by treatment with purified *anti*-OXM IgG to neutralize the effects of endogenous OXM in wild-type (*Figure 6E*) and in *Glp1r*$^{-/-}$ mice (*Figure 6F*). Importantly, the effects of refeeding on insulin, GLP-1, and GCG plasma levels were not affected by treatment with *anti*-OXM IgG, suggesting that these peptides are not involved in the activation of food-induced hepatic *Per* expression (*Figure 6G–I*).

We next tested if OXM signaling regulates food-mediated phase resetting of the liver clock using *Per2::LUC* slice preparations. *Per2::LUC* mice were starved overnight (ZT12-24/0) and treated with either IgG or *anti*-OXM antibodies prior to refeeding or extended starving. Animals were sacrificed at ZT4 and liver slices were prepared to determine luciferase rhythm phases. After a 12-hr fast, food intake caused a significant phase delay of PER2::LUC activity in liver slices and this effect was attenuated by OXM neutralization (*Figure 6J,K*). No effect on PER2::LUC phase was seen after *anti*-OXM treatment alone (*Figure 6K*).

Together, these data suggest that elevated OXM levels in response to food intake affect hepatic clock gene expression. Neutralization of endogenous OXM signaling inhibits food-induced clock gene induction and rhythm shifts, suggesting that OXM may act as a metabolic synchronizer of hepatic clocks.

## Discussion

Food intake resets circadian clocks in peripheral tissues. In consequence, eating during the normal

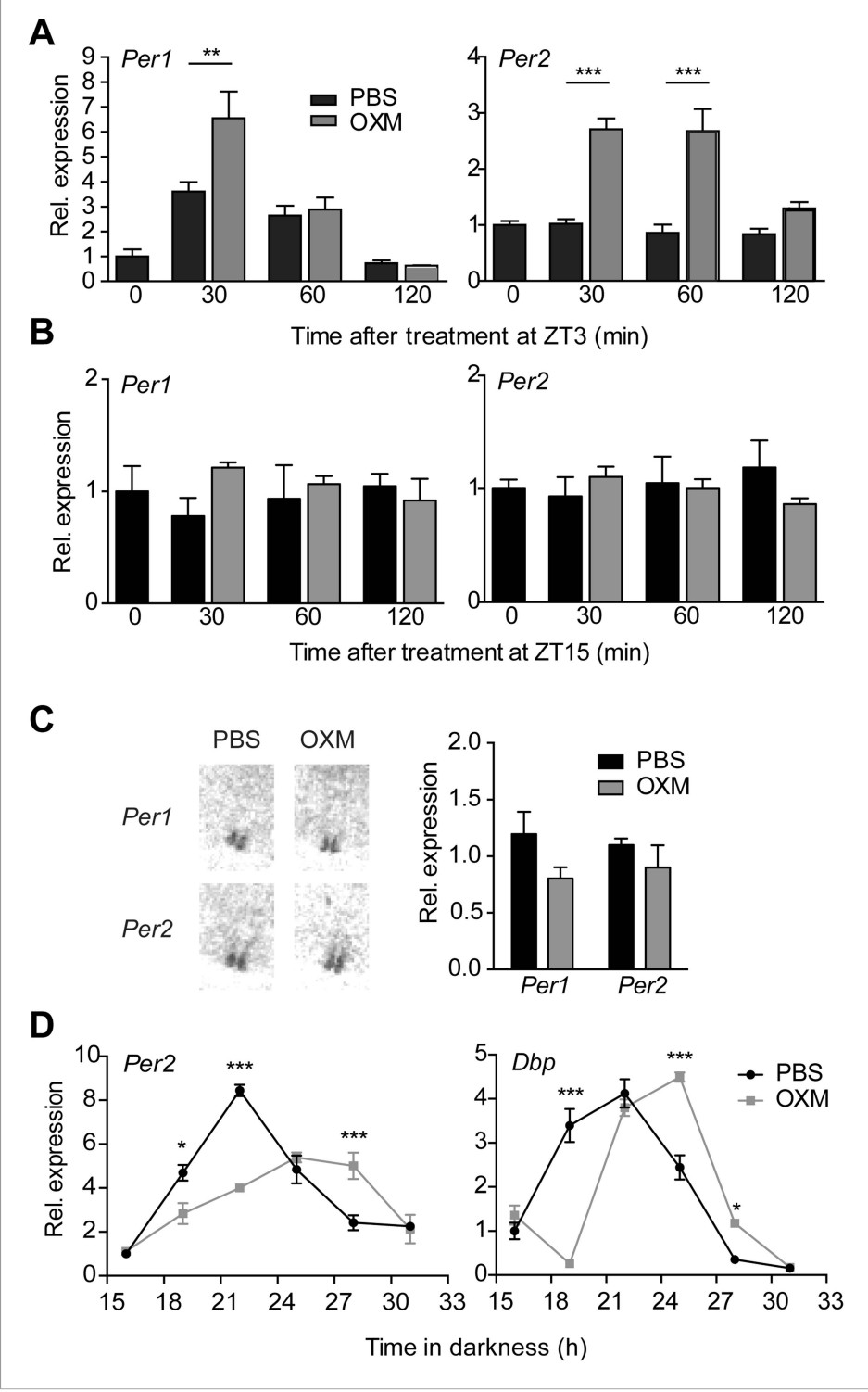

**Figure 4**. OXM treatment induces *Per1/2* expression and resets the liver circadian clock in vivo. (**A** and **B**) Hepatic *Per* gene expression after OXM (grey) or vehicle (PBS; black) *i.v.* injection at ZT3 (**A**) and ZT15 (**B**). ZT3: *Per1*: factor treatment $F_{(1, 24)} = 5.695$; time $F_{(2, 24)} = 34.74$; interaction $F_{(2, 24)} = 4.965$; *Per2*: factor treatment $F_{(1, 24)} = 64.84$; time $F_{(2, 24)} = 9.381$; interaction $F_{(2, 24)} = 6.915$. ZT15: *Per1*: factor treatment $F_{(1, 12)} = 1.096$; time $F_{(2, 12)} = 0.005$; interaction $F_{(2, 12)} = 1.367$; *Per2*: factor treatment $F_{(1, 24)} = 0.255$; time $F_{(2, 24)} = 0.001$; interaction $F_{(2, 24)} = 1.172$. (**C**) Suprachiasmatic nucleus (SCN) signal after in situ hybridization (ISH) of brain sections with $^{35}$S-labelled antisense probes for *Per1/2* 30 min after OXM/PBS treatment at ZT3 in the same animals used in (**A**). Left panel: representative

*Figure 4. continued on next page*

*Figure 4. Continued*

autoradiograph scans containing the SCN; right panel: quantification of the ISH. (**D**) Resetting of *Per2* and *Dbp* rhythms in livers of wild-type mice after an *i.p.* injection of either OXM (grey) or vehicle (PBS; black) after 12-hr darkness; *Per2*: factor treatment $F_{(1, 24)}$ = 5.531; time $F_{(5, 24)}$ = 46.37; interaction $F_{(5, 24)}$ = 18.71. *Dbp*: factor treatment $F_{(1, 24)}$ = 0.094; time $F_{(5, 24)}$ = 119.2; interaction $F_{(5, 24)}$ = 38.58. All data are presented as mean ± S.E.M. (n = 3–5). **A**, **B**, and **D**: two-way ANOVA with Bonferroni post-test: *$p < 0.05$, **$p < 0.01$; ***$p < 0.001$; **C**: Mann–Whitney test.

rhythms, as well as genes involved in hepatic carbohydrate regulation, thus impinging on hepatic energy metabolism (*Figures 4, 5*). In contrast, no significant effect of OXM treatment was observed at the level of the SCN pacemaker (*Figure 4*). In mice, food intake during day (i.e., the normal rest time) uncouples the liver clock from that in the SCN (*Lamia et al., 2008*), leading to a state of internal circadian desynchrony that is associated with elevated body weight and other metabolic impairments (*Arble et al., 2009*; *Hatori et al., 2012*). In the liver, around 10% of the transcriptome is under circadian regulation including many genes involved in energy metabolism (*Akhtar et al., 2002*; *Miller et al., 2007*). Genetic ablation of the liver clock abolishes the circadian rhythms of several glucose regulatory genes and results in a perturbed diurnal profile of blood glucose (*Kornmann et al., 2001*; *Lamia et al., 2008*).

Our experiments suggest that OXM signaling involves activation of GCGR (*Figure 2*). This is puzzling given that OXM displays a much greater capacity for clock gene induction in liver slices than GCG itself, despite having a lower binding affinity for GCGR (*LeSauter et al., 2009*). While we cannot conclusively answer this question at the moment, there are two scenarios that might explain our observations. First, GCGR may not be the only receptor involved in OXM-mediated clock resetting. While our data suggest that GLP1R does not play a role in this context (*Figures 2, 3*), an additional OXM receptor has been suggested (*Baldissera et al., 1988*; *Sun et al., 2015*), the activation of which may interact with GCGR downstream signaling. Second, an additional signal may be involved that acts synergistically with OXM to activate hepatic *Per* transcription. One such candidate could be insulin, which is similarly elevated after food intake and has previously been suggested to affect liver clock rhythms (*Yamajuku et al., 2012*; *Chaves et al., 2014*; *Sun et al., 2015*). In response to food intake OXM levels go up, while GCG plasma concentrations are reduced (*Figure 6*), suggesting that postprandial GCGR activation does not depend on GCG itself. Interestingly, GCG/GCGR signaling has recently been implicated in the regulation of hepatic *Bmal1* expression in response to prolonged starvation (*Sun et al., 2015*). Sun et al. show that fasting-induced GCG signaling activates *Bmal1* transcription via CREB/CRTC2 during the night. As we did not observe any acute effects of OXM treatment on *Bmal1* transcription (*Figure 3*), these data suggest that GCGR signaling may have different clock targets depending on the time of activation. GCGR activation is known to induce protein kinase A-mediated nuclear translocation and DNA-binding of phosphorylated CREB on target genes (*Gonzalez and Montminy, 1989*; *Dalle et al., 2004*). In line with this, we show OXM-induced binding of CREB to *Per1* promoter *CRE* motifs and induction of *Per* clock gene expression (*Figures 2–4*). Similar phase-dependencies were observed for *Per* induction as were seen for clock shifting (compare *Figures 1, 3, 4*). For various tissues—including the SCN—it has been shown that resetting of the circadian clock involves acute up-regulation of *Per1* and *Per2* (*Dunlap, 1999*; *Lowrey and Takahashi, 2004*). *Per* induction was observed after DEX treatment or serum shock in fibroblast and hepatoma cells (*Balsalobre et al., 1998*, *2000*). Light pulses given during the dark phase induce CREB activation and *Per* expression in the SCN (*Albrecht et al., 1997*; *Yan and Silver, 2002*). In line with this, we showed *Per1* and *Per2* induction after OXM treatment. Of note, *Per* induction in liver slices was weaker than in vivo, suggesting that additional mechanisms may amplify OXM effects in intact animals. Along the same line, OXM-mediated *Per* induction in vivo appeared to be slightly faster than in slices (*Figures 3, 4*). Importantly, peripheral treatment with OXM also altered the diurnal expression rhythms of genes involved in regulating liver carbohydrate metabolism which may underline the metabolic effects of daytime feeding in rodents (*Figure 5*). Similar to the OXM effects on clock gene expression (*Figures 3, 4*), refeeding acutely induces hepatic *Per1/2* expression, which has been proposed as an integral part of the food-driven clock-resetting mechanism (*Oike et al., 2011*; *Tahara et al., 2011*). In our study, we demonstrated that food intake after overnight fasting stimulated OXM

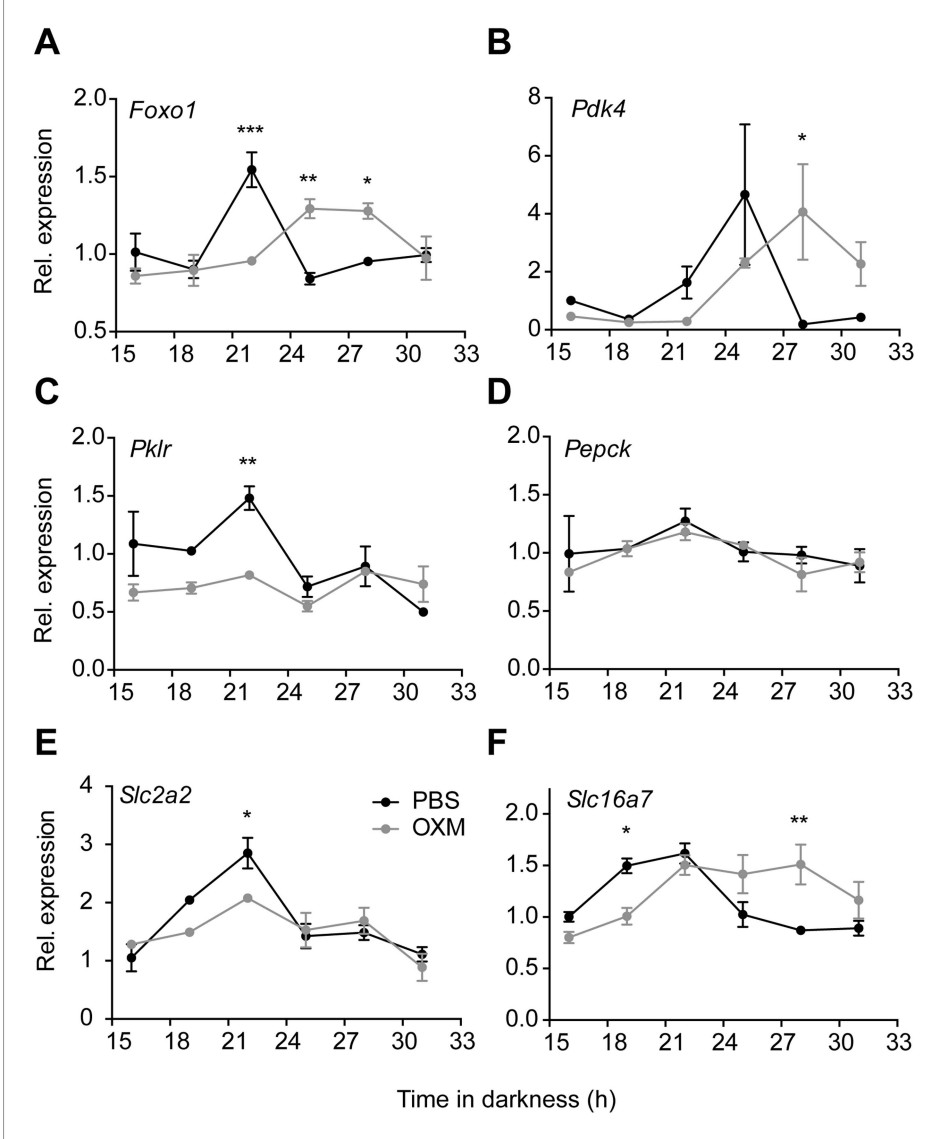

**Figure 5**. OXM treatment modulates diurnal expression profile of hepatic genes involved in liver carbohydrate metabolism. (**A–F**) Relative gene expression of *Foxo1* (**A**; factor treatment $F_{(1, 24)} = 0.001$, time $F_{(5, 24)} = 5.547$, interaction $F_{(5, 24)} = 11.13$), *Pdk4* (**B**; factor treatment $F_{(1, 24)} = 0.197$, time $F_{(5, 24)} = 3.35$, interaction $F_{(5, 24)} = 3.247$), *Pklr* (**C**; factor treatment $F_{(1, 24)} = 11.63$, time $F_{(5, 24)} = 5.61$, interaction $F_{(5, 24)} = 3.61$), *Pepck* (**D**; factor treatment $F_{(1, 24)} = 0.574$, time $F_{(5, 24)} = 2.043$, interaction $F_{(5, 24)} = 0.299$), the glucose transporter *Slc2a2* (**E**; factor treatment $F_{(1, 24)} = 2.582$, time $F_{(5, 24)} = 15.98$, interaction $F_{(5, 24)} = 2.642$) and the pyruvate transporter *Slc16a7* (**F**; factor treatment $F_{(1, 24)} = 1.539$, time $F_{(5, 24)} = 7.472$, interaction $F_{(5, 24)} = 6.586$) after *i.p.* administration of either OXM (grey) or vehicle (PBS; black) after 12 hr in darkness. Data are presented as mean ± S.E.M. (n = 4). Two-way ANOVA with Bonferroni post-test: *p < 0.05, **p < 0.01; ***p < 0.001.

secretion and led to hepatic up-regulation of *Per* expression, which was blocked by OXM neutralization in the circulation. Accordingly, liver slice cultures from re-fed *Per2::LUC* reporter mice showed food-dependent phase delays and these effects were attenuated by OXM neutralization (*Figure 6*). OXM neutralization does not completely inhibit the effects of refeeding on *Per* induction and liver clock resetting. This might be due to incomplete neutralization of OXM or the contribution of other food-induced factors as discussed above. Of note, while our data do not provide evidence for a direct involvement of GLP-1(R), it has been suggested that postprandial GLP-1 signaling may indirectly affect liver clock phase (*Panjwani et al., 2013*).

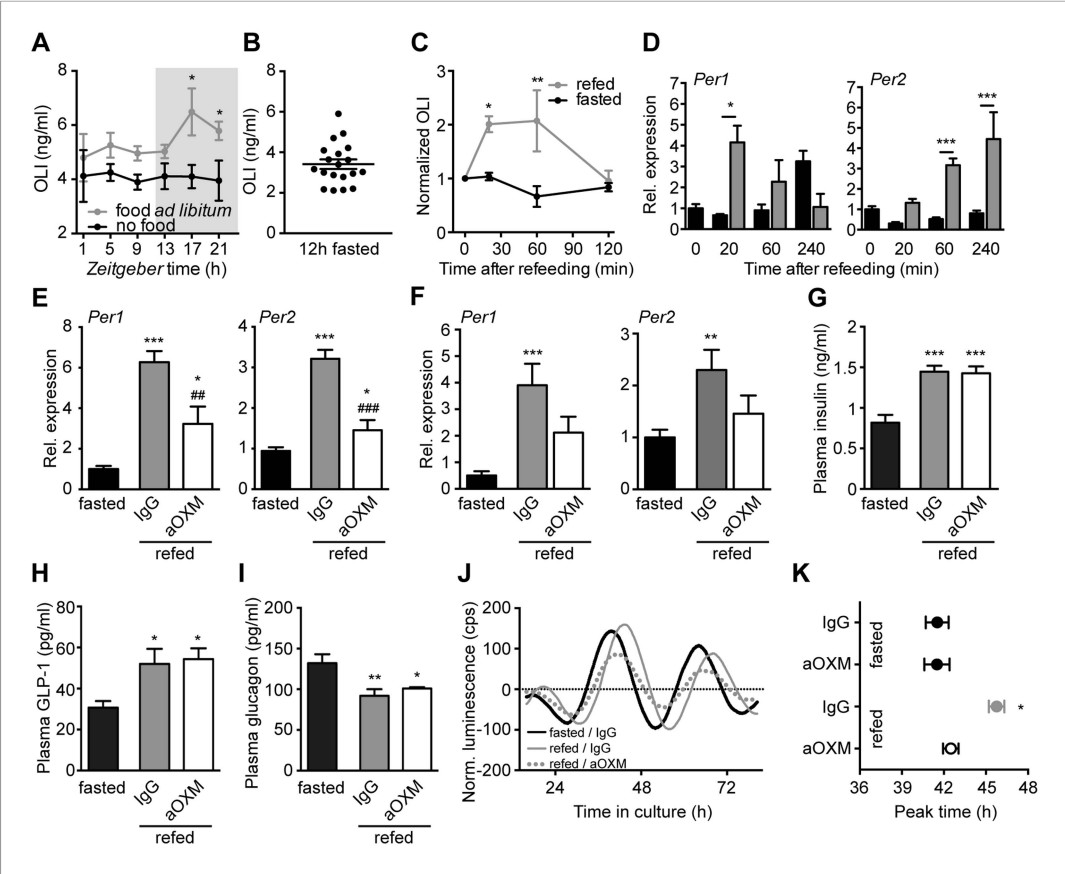

**Figure 6**. Endogenous OXM signaling regulates food intake-mediated resetting of the liver circadian clock. (**A**) Plasma oxyntomodulin-like immunoreactivity (OLI) diurnal profiles under ad libitum food and fasting conditions. Data are presented as mean ± S.E.M (n = 6); factor time $F_{(5, 60)}$ = 0.628, feeding condition $F_{(1, 60)}$ = 15.37, interaction $F_{(5, 60)}$ = 0.638. Grey shading indicates the dark phase. (**B**) OLI levels show high individual variations in mice after 12 hr of food deprivation (ZT13-1). (**C**) Plasma OLI (normalized to individual fasting levels) after refeeding (grey line) or under continuous starving (black line); factor time $F_{(3, 21)}$ = 3.544, feeding condition $F_{(1, 21)}$ = 15.82, interaction $F_{(3, 21)}$ = 4.717. (**D**) Liver *Per1/2* induction following fasting-refeeding determined by qPCR; *Per1*: factor time $F_{(3, 25)}$ = 0.454, feeding condition $F_{(1, 25)}$ = 1.376, interaction $F_{(3, 25)}$ = 4.453; *Per2*: factor time $F_{(3, 25)}$ = 6.938, feeding condition $F_{(1, 25)}$ = 38.48, interaction $F_{(3, 25)}$ = 3.767. (**E**) WT and (**F**) *Glp1r$^{-/-}$* liver *Per1/2* expression after fasting-refeeding with control IgG injection (grey) or OXM immuno-neutralization by *anti*-OXM IgG (aOXM) injection at ZT0; WT: *Per1*: $F_{(2, 12)}$ = 71.76, *Per2*: $F_{(2, 12)}$ = 47.41; *Glp1r$^{-/-}$*: *Per1* $F_{(2, 12)}$ = 11.51, *Per2* $F_{(2, 12)}$ = 5.585. (**G–I**) Treatment with *anti*-OXM IgG does not affect postprandial regulation of insulin, GLP-1, and GCG. Plasma levels of insulin (**G**; $F_{(2, 12)}$ = 17.44), GLP-1 (**H**; $F_{(2, 12)}$ = 5.563), and GCG (**I**; $F_{(2, 12)}$ = 7.128) after fasting-refeeding with control IgG injection (grey) or OXM immuno-neutralization by *anti*-OXM IgG (aOXM) treatment at ZT0. One- (**E–I**) or two-way ANOVA (**A**, **C**, **D**) with Bonferroni post-test: *p < 0.05; **p < 0.01; ***p < 0.001 against fasted; ##p < 0.01; ###p < 0.001 against IgG. Data are presented as mean ± S.E.M (n = 5). (**J** and **K**) Liver PER2::LUC rhythms after fasting-refeeding with control IgG or αOXM administration. (**J**) Representative luminescence traces. (**K**) Comparison of phases (second peak in culture) after refeeding and/or *anti*-OXM treatment (Data are presented as mean ± S.E.M (n = 4 mice per condition, an average of 3 slice preparations of each mouse were used); two-way ANOVA with Bonferroni post-test: *p < 0.05 against fasted; factor treatment $F_{(1, 12)}$ = 5.127, feeding condition $F_{(1, 12)}$ = 13.02, interaction $F_{(1, 12)}$ = 5.044).

The following source data and figure supplements are available for figure 6:

**Source data 1**. Primer sequences for PCR reactions.

**Figure supplement 1**. Time course of OLI plasma levels after OXM injection.

Our data indicate an involvement of OXM signaling in food-driven resetting of the liver circadian clock. Similar to what was previously reported in humans (*Mayo et al., 2003*), we detected elevated OLI levels in fed as compared to starved mice and in response to acute food intake (*Figure 6*). The diurnal variability of OLI under ad libitum feeding conditions was moderate, but fasting resulted in an approximate 20% reduction of diurnal OLI secretion. Together this indicates that small meals as consumed during the inactive phase affect OXM secretion, but that substantial food intake is necessary to acutely increase OXM secretion to an extent sufficient to affect the liver clock. Of note, interpretation of plasma OXM blood levels is difficult as many OXM assays exhibit some degree of cross-reactivity with GCG and the closely related glicentin. Glicentin differs from OXM only by a 32-amino acid N-terminal extension (IP-1; *Figure 1*) and is released from the gut after food intake at approximately similar levels (*Blache et al., 1988*; *Tang-Christensen et al., 2001*), thus we cannot exclude the possibility that glicentin may exhibit actions that overlap with those described for OXM.

In summary, we show that food intake induces OLI blood levels and hepatic clock resetting in mice. OXM treatment mimics food-mediated clock resetting in slices and in vivo in a time of day-dependent manner. Food is a major regulator of hepatic transcriptome rhythms (*Vollmers et al., 2009*). A role for metabolic hormones, for example, insulin, GCG, or glucocorticoids, in the regulation of peripheral clocks has been suggested (*Balsalobre et al., 2000*; *Tahara et al., 2011*; *Sun et al., 2015*). However, the mechanisms of food-related synchronization of peripheral clocks and their uncoupling from the SCN under time-restricted food intake rhythms are still poorly understood. Our data suggest that OXM—most likely in concert with other gut- and pancreas-derived hunger- or satiety-signaling peptides—is involved in the detrimental effects of mistimed food intake on metabolic homeostasis (*Arble et al., 2009*; *Albrecht, 2012*; *Hatori et al., 2012*). While interfering with insulin signaling may be clinically problematic because of its potential deleterious effects on glycemia, chronotherapeutic targeting of peripheral incretin signaling may provide an alternative therapeutic strategy against metabolic disorders arising from circadian strain as observed in shift workers or during jetlag.

## Materials and methods

### Animal strains

All animal experiments were ethically assessed and licensed by the Office of Consumer Protection and Food Safety of the State of Lower Saxony and in accordance with the German Law of Animal Welfare (license nos. V312-7224.122-4 and 33.12-42502-04-12/0893). For all experiments adult wild-type mice male C57BL/6J (8–24 weeks old) were used. If not stated otherwise, mice were provided with food and water ad libitum. To investigate the effect of OXM on gene expression by qPCR or ISH, mice were peripherally treated with OXM (*i.v.* 4 µg/mouse; *i.p.* 25 µg/mouse) or vehicle (PBS) at ZT3, ZT15, or 12 hr after light-off. For luminescence measurements adult heterozygous males *Per2::LUC* (*Yoo et al., 2004*) and *Per2::LUC* x *Glp1r*$^{-/-}$ were used. *Glp1r*$^{-/-}$ mice were maintained on a C57B/6J background (*Scrocchi et al., 1996*). All mice were exposed to a 12-hr: 12-hr light–dark cycle with 100 lux in the light phase (LD12:12). Animals were sacrificed at indicated time points by cervical dislocation. Animals euthanized during the dark phase were handled under red light and eyes were removed before dissection. All tissue samples were collected and immediately snap-frozen on dry ice or liquid nitrogen. For long-term storage tissues were kept at −80°C.

### Refeeding experiments

In order to test *Per* induction and phase shifts after refeeding, mice were fasted for one night (13 hr, or from ZT12—ZT1) and either food deprived until decapitation or refed at ZT1. The refed mice received either *anti*-OXM rabbit IgG (*i.p.* 50 µg/mouse; T-4800; Bachem, Bubendorf, Switzerland) or control IgG from unimmunized rabbit serum (I5006; Sigma-Aldrich, Seelze, Germany) at ZT1 when food was returned.

### Liver slice cultures and peptide treatments

Luminescence was measured from cultured liver slices of heterozygous *Per2::LUC* mice as described previously (*Yoo et al., 2004*) modified to include the use of culture plate inserts (Millipore, Billerica, MA). Briefly, the median lobe of the liver was isolated and 300-µm slices were prepared using a vibratome (Campden Instruments, Loughborough, UK). The slices were immediately placed onto a culture plate insert in 35-mm petri dishes filled with 1-ml culture medium (D-MEM with high glucose,

w/o L-glutamine and phenol red; Life Technologies, Darmstadt, Germany) supplemented with 3 mM sodium carbonate (Sigma–Aldrich), 10 mM HEPES buffer, 2 mM L-glutamine, 2% B-27 supplement, 25 U/ml penicillin/streptomycin and 0.1 mM D-luciferin (all Life Technologies). Luminescence was measured in a luminometer (Actimetrics, Evanston, IL) at 32.5°C with 5% $CO_2$. Analyses were performed using the LumiCycle analysis (Actimetrics) and Prism software packages (GraphPad, La Jolla, CA). PER2:: LUC activity in slices closely follows a sine wave shape. The intersection of the ascending cross-section of the sine wave with the x-axis was defined as 0°/360°, the peak as 90°. Degrees at the time point of treatment were calculated as follows: $T_p [°] = ((T_p [hsm] - P_{bt} [hsm]): P_{bt}) \times 360 + 90$ with $T_p$ = treatment phase; ° = degree; hsm = hours after start of measurement; $P_{bt}$ = peak before treatment. If the result was >360°, the value was subtracted by 360. Raw data were baseline subtracted with running averages of 24 hr. Peaks were defined as middle time point between two troughs of the sine wave. Period was determined as the time between peaks averaged over 2–3 consecutive cycles. For the duration of treatment, samples were maintained at 32.5°C to avoid resetting of clock gene expression rhythms due to temperature changes. Phase shifts were determined by comparing extrapolated peak times from sine wave fits before and after treatment. Unless otherwise stated, peptides used for experiments were dissolved in culture medium and administered at a final concentration of 450 pM.

## Quantitative real-time (qPCR)

Quantitative real-time PCR (qPCR) was performed with a CFX96 thermocycler system (Bio-Rad, Munich, Germany) with iQ-SYBR Green SuperMix (Bio-Rad). Relative quantification of expression levels by a modified ΔΔCT calculation was performed as described (*Pfaffl, 2001*). *ß-Actin* was used as a reference gene. Statistical analyses were performed using GraphPad Prism software. Circadian profiles of clock gene expression were normalized against the average values over all time points. Induction analyses were normalized against untreated conditions (0 min). PCR primer sequences are listed in *Figure 6—source data 1*.

## ISH

The *Per1* probe corresponds to nucleotides 1 to 619 (GenBank accession number AF022992) and *Per2* corresponds to nucleotides 229 to 768 of GenBank AF036893. PCR products had been cloned into *pCR II TOPO* vector using TOPO TA Cloning Kit (Life Technologies) (*Oster et al., 2002*). Linearization of vectors for in vitro transcription was done with EcoRI. $^{35}$S-UTP (PerkinElmer, Waltham, MA) labeled RNA probes were prepared using RNA Transcription Kit (Maxi Script Labeling Kit, Life Technologies) with T7 or T3 RNA polymerases according to the manufacturer's protocol. 10-μm cryosections were cut using a Leica CM3050 cryostat. Cryosections were fixed in 4% paraformaldehyde, acetylated in acetic anhydride and dehydrated with ethanol. Hybridization was performed over night at 55–58°C. Autoradiographs were analyzed by densitometry (Bio-Rad GS-800) using QuantityOne software (Bio-Rad). Three sections per brain were used and background values were calculated from adjacent tissue areas on the same slide for each section. Measurements from different animals/experiments were compared for statistical analysis using GraphPad Prism (GraphPad).

## ChIP

Liver slices were homogenized and immediately cross linked with 1% formaldehyde. Chromatin was sonicated for 15-s on/20-s off cycles for 22 min using a Bioruptor sonicator (Diagenode, Denville, NJ). Samples were incubated overnight at 4°C with CREB antibody (ab31387, Abcam, Cambridge, UK). After clearing, samples were incubated with A/G agarose beads (Thermo Scientific, Braunschweig, Germany) for 1 hr at 4°C followed by repetitive washings. After boiling for 10 min in 10% Chelex (Bio-Rad) with Proteinase K (150 mg/ml), samples were spun down and DNA-containing supernatant was collected for PCR. qPCR was performed as described above, and values were normalized to percentage of input. Primer sequences were: 5′-CAGCTGCCTCGCCCCGCCTC-3′/5′-CCCAAGCAGCCATTGCTCGC-3′ (Per1 CRE) and 5′-CCCCGCAGTCCTACGGTGCTG-3′/5′-AAGCCCCCAAACAACTGAAGGT-3′ (500 bp downstream sequence).

## Hormone measurements

Blood collection for radioimmunoassay (RIA) was performed at ZT1 after 12 hr fast. Mice were allowed to recover for 3 days, then fasted again followed by treatments (with or without refeeding). Blood was collected from the tail vein at 0 min, 20 min, 60 min, and 120 min after treatment. Plasma concentrations of OLI were determined by RIA (Phoenix Pharmaceutics, Karlsruhe, DE) according to

manufacturer's protocol modified to use a 50% reaction volume. GLP-1 (EZGLP-1T-36K, Millipore, Darmstadt, Germany), insulin (Catalog# 90080, CrystalChem, Downers Grove, IL), and GCG plasma levels (EZGLU-30K, Millipore) were determined by ELISA according to the manufacturers' protocols.

## Statistics

Data were analyzed with GraphPad Prism (GraphPad). Mann–Whitney tests were used for simple comparisons. For dose responses one-way ANOVAs and for two-factor comparisons two-way ANOVAs with Bonferroni post-tests were used. A p-value of less than 0.05 was considered significant.

## Acknowledgements

The authors thank Bernard Thorens, University of Lausanne for indirectly providing $Glp1r^{-/-}$ mice, Nadine Naujokat and Christin Helbig for technical assistance, and David K Welsh, UC San Diego, and Gregor Eichele, Max Planck Institute for Biophysical Chemistry Göttingen, for critical comments on the manuscript.

## Additional information

### Funding

| Funder | Grant reference | Author |
|---|---|---|
| Volkswagen Foundation | Lichtenberg Fellowship | Henrik Oster |
| Deutsche Forschungsgemeinschaft (DFG) | TR-SFB134 | Henrik Oster |
| Canada Research Chairs (Chaires de recherche du Canada) | | Daniel J Drucker |
| Canadian Institutes of Health Research (Instituts de recherche en santé du Canada) | 93749 | Daniel J Drucker |
| Novo Nordisk | Chair of Incretin Biology | Daniel J Drucker |

The funders had no role in study design, data collection and interpretation, or the decision to submit the work for publication.

### Author contributions

DL, AHT, Acquisition of data, Analysis and interpretation of data, Drafting or revising the article; AL, CEK, JLB, Acquisition of data, Analysis and interpretation of data; DJD, Drafting or revising the article, Contributed unpublished essential data or reagents; HO, Conception and design, Acquisition of data, Analysis and interpretation of data, Drafting or revising the article

### Ethics

Animal experimentation: This study was performed in strict accordance with the German law for animal welfare (TierschGes). All animals were handled according to approved institutional animal care and use committee protocols of the Max Planck Institutes Göttingen and the University of Lübeck. The protocol was approved by the ethical committees of the Niedersächsisches Amt für Verbraucherschutz und Lebensmittelsicherheit (LAVES) and the Ministerium für Energiewende, Landwirtschaft, Umwelt und ländliche Räume (MELUR; license numbers V312-7224.122-4 and 33.12-42502-04-12/0893). Every effort was made to minimize suffering.

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
