## [Decision Letter]

Thank you for sending your work entitled “Oxyntomodulin regulates resetting of the liver circadian clock by food” for consideration at *eLife*. Your article has been favorably evaluated by Janet Rossant (Senior editor) and 3 reviewers, one of whom, Louis Ptacek, is a member of our Board of Reviewing Editors.

The Reviewing editor and the other reviewers discussed their comments before we reached this decision, and the Reviewing editor has assembled the following comments to help you prepare a revised submission.

The findings in this paper strongly point to a physiological role of OXM in the regulation of the liver circadian clock that arises following the ingestion of food. This study complements earlier work by other groups that examine how corticosterone or body temperature can synchronize clocks in peripheral tissues. OXM has not been implicated in this process and this study demonstrates a new role for this neuropeptide. Moreover, the findings also point to unusual receptor signaling mechanisms as underpinning the actions of OXM in the liver.

Minor comments:

1) How do the doses of OXM used in this study compare to what is known regarding the physiological concentrations of OXM?

2) Mice generally eat most during the lights-off phase, but will occasionally spontaneously eat small meals and water during the lights-on phase. Does this mean then that the release of a critical liver resetting concentration of OXM is only achieved when substantially more food is consumed during the day than is typical?

3) Are the effects of OXM on Per2 in the liver always indicative of induction? There seems to be some variation from one experiment to another or possible alternative interpretations.

4) A statement in the Methods on how the results were statistically analysed would be helpful, e.g. what statistics software package was used, significance levels used etc. Also, why not report F-values and Deg. of Freedom in the figure captions?

5) What is known about the cycling of OXM transcript in the gut? Does this vary circadianly in the absence of food intake?

6) It would be informative to include representative examples of bioluminescence rhythms from the liver slices, showing the phase shift effect of OXM. This would make it clearer to a reader not familiar with such experiments.

---

## [Author Response]

1) How do the doses of OXM used in this study compare to what is known regarding the physiological concentrations of OXM?

We have added a new supplemental figure (Figure 6—figure supplement 1) in which we measure blood OLI levels after i.v. and i.p. injection as used in this paper. When one compares these to the refeeding data (Figure 6), it becomes clear that the OXM concentrations after injection are around 2-3 fold higher than after re-feeding. We added this information into the subsection of the Results headed “Food intake-mediated OXM induction and liver clock resetting”.

2) Mice generally eat most during the lights-off phase, but will occasionally spontaneously eat small meals and water during the lights-on phase. Does this mean then that the release of a critical liver resetting concentration of OXM is only achieved when substantially more food is consumed during the day than is typical?

We now include a new panel in Figure 6 (revised Figure 6) where we compare blood OLI levels between mice under ad libitum feeding conditions and fasted animals. Overall OLI levels are 20% higher in fed animals (p < 0.001 for feeding condition in a 2-way ANOVA), and the difference is strongest during the lights-off phase. This suggests that even small meals as consumed during the light phase can affect OLI, but the overall variation is too high for this effect to reach significance at any single time point during the lights-on phase. We discuss this in the Discussion section.

*3) Are the effects of OXM on Per2 in the liver always indicative of induction? There seems to be some variation from one experiment to another or possible alternative interpretations*.

In our hands *Per2* induction was consistently observed after OXM treatment. However, without longitudinal measurements (e.g. by using in vivo PER2::LUC recordings as introduced by Saini et al. Genes Dev 2013 Jul 1;27(13):1526-36) occasional non-responders cannot be excluded.

*4) A statement in the Methods on how the results were statistically analysed would be helpful, e.g. what statistics software package was used*, *significance levels used etc. Also, why not report F-values and Deg. of Freedom in the figure captions?*

We have added this information (Methods section and in the figure captions).

5) What is known about the cycling of OXM transcript in the gut? Does this vary circadianly in the absence of food intake?

We have measured *Gcg* mRNA rhythms (as the precursor mRNA of OXM) in the gut under fasting conditions (see Figure 7) and blood OLI levels under ad libitum food and fasting conditions (revised Figure 6). mRNA and peptide levels are non-rhythmic under fasting conditions, while under ad libitum feeding OLI is elevated during the dark phase when mice are active and eat most of their food. Since *Gcg* also gives rise to other peptides besides OXM, we believe that the mRNA data are not very informative for our story and, thus, only mention these data briefly in the main text (in the subsection headed “Food intake-mediated OXM induction and liver clock resetting”)

Author response image 1.Circadian mRNA expression of *Gcg* and *Dbp* in the gut of fasted mice. Animals were released into constant darkness and samples were collected at 4 time points under fasting conditions on day 3. While *Dbp* shows a robust circadian rhythm, *Gcg* mRNA is expressed at comparable levels throughout the circadian cycle. Data are mean ± SEM (n = 4).**DOI:**
http://dx.doi.org/10.7554/eLife.06253.016

*6) It would be informative to include representative examples of bioluminescence rhythms from the liver slices, showing the phase shift effect of OXM. This would make it clearer to a reader not familiar with such experiments*.

We have added representative bioluminescence traces to Figure 1 to exemplify the data obtained from this type of resetting experiments.